# Influence of Monosodium Glutamate and Its Substitutes on Sensory Characteristics and Consumer Perceptions of Chicken Soup

**DOI:** 10.3390/foods8020071

**Published:** 2019-02-14

**Authors:** Shangci Wang, Shaokang Zhang, Koushik Adhikari

**Affiliations:** Department of Food Science and Technology, University of Georgia-Griffin Campus, 1109 Experiment Street, Griffin, GA 30223, USA; wsc727@uga.edu (S.W.); zskzsk@uga.edu (S.Z.)

**Keywords:** monosodium glutamate (MSG), MSG substitutes, food label, chicken soup

## Abstract

Soup manufacturers are removing monosodium glutamate (MSG) to meet consumer demand for natural ingredients. This research investigated the influence of MSG and its substitutes (yeast extract: YE; mushroom concentrate: MC; tomato concentrate: TC) on clear chicken soup with 0.4% sodium chloride (salt) by comparing sensory attributes and consumer acceptability among each other, and to a chicken soup sample containing 0.5% salt (Salt 0.5%). The soup with 0.4% salt without enhancers was designated as the control. Corresponding list of ingredients was also presented to consumers to study the effects on consumer expectations about chicken soup products. Our results showed that MSG and its substitutes significantly (*P* < 0.05) enhanced the sensory properties of chicken soup. These flavor enhancers also achieved statistically same or stronger improvement in overall flavor, meaty flavor, chicken flavor and umami taste when compared to Salt 0.5% sample. Consumers significantly preferred MSG 0.1%, YE 0.025%, and Salt 0.5% samples than others. Compared to MC and TC samples, less consumers perceived MSG and YE samples as “free of artificial” and “natural” with lower consumption interest. Claims about artificial/natural ingredients were attractive selling points for chicken soups, but good sensory appealing was the most important attribute linearly affecting consumer satisfactions.

## 1. Introduction

Soup is an important food product category in the market. According to Grand View Research, Inc., the US soup market size is expected to reach ~7.7 billion US dollars by 2025 [1]. Chicken soup has been known for centuries as home remedy for cold and flu, possibly due to its hot temperature and anti-inflammatory effects [2,3]. In addition, chicken soup can help with weight control and reduction because of its high water content. There are several trends that drive the growth of soup industry, including demands for natural and fresh ingredients and rising awareness of healthy food choice. To further fit the natural and healthy portfolio, several major soup manufacturers have announced to move away from using artificial ingredients and flavors in their products. Monosodium glutamate (MSG) is one such ingredient that has been controversial for decades. It is one of the ingredients that some companies have committed to remove from products.

MSG is a flavor enhancer commonly added to processed food products like chicken soup to boost the palatability. Its remarkable effects on the sensory appeal have been proven in various studies [4,5,6]. Removal of this ingredient is very likely to cause a reduced consumer acceptability. Using MSG substitute is a promising approach to compensate for the sensory satisfaction loss caused by MSG elimination. The flavor enhancement effect of MSG is mainly from glutamate which contributes to umami or savory taste sensation. Besides glutamate, there are several other umami eliciting components such as aspartate and 5’-ribonicleotides. Among nucleotides, inosinate (IMP) and guanylate (GMP) significantly contribute to flavor and taste enhancement. Theoretically, substances that are naturally rich in umami components have the potential to replace MSG in food products. Wang and Adhikari found that consumers preferred natural extracts such as yeast extract, mushroom extract, and tomato extract as MSG substitute in chicken soup products [7]. Yeast typically has around 7–12% of RNA content which makes it a good source to produce nucleotide-rich ingredients [8]. Previous studies found that compared to MSG, yeast extract (YE) had stronger salty and umami taste at the same usage level and better improved the tastes of meat samples [5,9]. Mushroom is another nucleotide-rich material. For instance, GMP is originally isolated from shiitake mushroom. Ground mushroom can substitute 80% beef in taco blend with enhanced overall flavor and umami taste [10]. Tomato is very rich in glutamate. In contrast to MSG, tomato puree can better enhance the sweet, salty, and sour taste of minced beef but suppress the beefy flavor [5]. In addition, processing like drying can remarkably boost the glutamate content in tomato and mushroom and generate more GMP in mushroom [11]. Currently, there is limited research comparing the enhancement effects of MSG with these natural extracts in food products.

Excessive sodium intake is major health concern in the US, which makes sodium reduction almost a necessity in soup manufacturing. Unfortunately, reducing table salt usage always results in consumer complaints of bland taste and flavor. Application of MSG is a common approach to improve palatability of low sodium soups. Given the capability of salty taste enhancement, MSG substitute may also be able to increase the sensory appeal of soup with reduced salt content. Previous study indicated that 1% and 2% yeast extract successfully enhanced the taste of fermented sausage that had 25% NaCl replaced by KCl [12]. Ground mushroom has also been reported to improve the flavor of taco blend with 25% sodium reduction but failed to mitigate the reduced salty taste [10]. To replace MSG, it is necessary to conduct more research to compare the performance between MSG and its alternatives in salt-reduced food matrix.

Consumer’s food choice is a tradeoff among sensory and non-sensory factors [13]. As the rising health awareness, consumers use food labels to obtain the information they require. Information on food packaging plays a crucial role in consumer purchasing decisions. Consumers are likely to imagine the tastes and expect the health benefits of the product based on the important cues from packaging, which might further influence their hedonic and sensory perceptions [14]. Bandara et al. reported that MSG content affected the purchase intent of 21% of the respondents to a large extent [15]. However, Prescott and Young noted that information about MSG usage did not influence consumer liking and natural taste perception about chicken soups [16]. Thus, it would be interesting to investigate how the information about MSG and its substitute usage affects consumer expectations and perceptions about chicken soup products. Besides ingredient choice, there are several other claims that commonly appear on packaging of chicken soup products. However, it is unclear how these features influence consumer satisfactions. Kano analysis is a useful consumer research tool that is widely practiced in industries to classify and prioritize product features based on their influences on customer satisfaction.

This work applied MSG and its substitutes in chicken soup added with 0.4% NaCl to (1) compare their influences on sensory attributes and consumer acceptability of chicken soup and (2) investigate whether samples added with these flavor enhancers could achieve the sensory appeal of the chicken soup added with 0.5% salt. This study also aimed to analyze how information about MSG and its replacement usage affects consumer expectations about chicken soup products. In addition, Kano analysis was applied to identify how different features of chicken soup products influence consumer satisfactions about chicken soups.

## 2. Materials and Methods

### 2.1. Soup Preparation

The chicken soup stock was prepared weekly by boiling 1 kg drumsticks (separated into skins, flesh, and bones) in 3.5 L water in Crock-Pot with locking lid (Sunbeam Products, Inc., Boca Raton, FL, USA). Vegetables and spice (0.1 kg carrot, 0.1 kg yellow onion, 0.05 kg celery stalk, and 0.1 g bay leaf) were put on a four-layer cheese clothes in a tightly closed Crock-Pot. The stock was cooked at high heat for five hours and at low heat for three hours. Then the cheese clothes were immediately pulled from Crock-Pot and the stock was filtered through a four-layer cheese clothes again to generate a clear appearance. To balance the possible variations among the Crock-Pots, a completely randomized process was utilized to transfer the stock from Crock-Pot to glass jars: about 100–110 mL of stock was taken from individual Crock-Pot to fill up one 1.89 L glass jar. The jars were tightly sealed, cooled to room temperature, and stored at 4 °C for ~12 hours to remove top fat. The maximum storage of stock was four days at 4 °C. The chicken soup samples were prepared two hours before each sensory test. Table 1 presents the levels of NaCl and flavor enhancer in 1.5 L stock to make the soup. The usage level was selected based on the Equivalent Umami Concentration (EUC) of umami substances of each ingredient and their sensory enhancement effects on the chicken soup sample (data submitted elsewhere). The ingredients were added to 1.5 L of the stock at the beginning of the soup preparation and completely dissolved in the stock by string. Chicken soup was then heated at high heat in tightly closed Crock-Pot for one hour and 10 min and then kept at warm until the end of tests.

### 2.2. Degree of Difference from Control Testing

Approval from UGA’s IRB (Project ID: STUDY00004396) was obtained before collecting the sensory data.

A degree of difference from control (DODC) test was carried out to find flavor differences in the test chicken soup samples in comparison to the control sample. A DODC test is a combination of discrimination and descriptive tests. Two samples are served together as in discrimination test (one control and one test sample) to examine if the samples are different. But it has another layer similar to descriptive analysis, where the source of differences is identified and the intensity of each difference is measured. References are not needed for a DODC test since control or reference sample is always served with the test sample. A total of six panelists who have more than 10 years of experience in descriptive analysis and one year experience in DODC tests were recruited for this study. Eight two-hour training sessions were conducted on the test samples to familiarize the panel with the major flavor attributes and to calibrate the panel’s performance. Eight impact attributes were identified through panel consensus, which were overall flavor, meaty flavor, chicken flavor, salty taste, umami taste, yeast flavor, mushroom flavor, and tomato flavor. A pair of samples were served together: one was the test sample labeled with 3-digit code and the other was marked as “Control”, which was the chicken soup with 0.4% salt. Panelists were required to compare the test sample against the “Control” for each attribute to evaluate the differences, if any. Each attribute was first rated on a comparison scale anchored at −1, 0 and 1, which represented “less than the Control”, “same as the Control”, and “greater than the Control” respectively. A 0 to 10-point intensity scale with 0.5 increments was then used to scale the degree of difference between the test sample and “Control”. The values of 2.5, 5, and 7.5 were labeled as “slight difference”, “moderate difference”, and “large difference”, respectively, on the intensity scale. About 60 mL of soup was served at 50 ± 2 °C in a 118-mL Styrofoam cups (Dart Container, Mason, MI, USA) wrapped with aluminum foil to avoid temperature loss during serving. The samples were tested in a random order and replicated twice. Six samples were evaluated against “Control” in each testing session. There was a two-minute break between two sets of samples to avoid panel fatigue. Unsalted Saltine crackers (Nabisco, East Hanover, NJ, USA) and deionized water were used as palate cleansers.

### 2.3. Consumer Testing of Chicken Soup

All the 6 samples were evaluated by 93 consumers who were recruited from an existing consumer database maintained at Sensory Evaluation and Consumer Lab, Department of Food Science and Technology, University of Georgia (Griffin Campus). All the consumers were recruited based on the following criteria: (1) 18 years of age and above, (2) 30% male, (3) no food allergies or intolerances, (4) shared at least equal responsibility of grocery shopping with other family members in the household, and (5) purchased and ate chicken soup products at least four times in the past year. Compusense^®^ Cloud (Compusense, Inc., Guelph, ON, Canada) was used for data collection.

#### 2.3.1. Taste Test

The consumer tests were carried out in partitioned booths under incandescent light at ~21 °C. About 60 mL of each chicken soup sample was served at ~50 ± 2 °C in a 118-mL Styrofoam cup. The samples were coded with three-digit random numbers and served in a sequential monadic order based on a completely randomized design. Unsalted Saltine crackers and distilled water were served as palate cleansers in-between samples. A nine-point hedonic scale (1—dislike extremely; 5—neither like nor dislike; 9—like extremely) was used for liking questions and a seven-point Just-about-right (JAR) scale (1—too weak; 4—JAR; 7—too strong) was used for color, overall flavor, salty, and savory (umami) tastes intensity questions.

#### 2.3.2. Expectation Test on Ingredient List

Five lists of ingredients for chicken soup samples were presented to consumers in a fixed sequential monadic order from Panel A to E (Table 2). According to the information on the ingredient list, consumers were firstly required to rate their opinions of “free of artificial ingredients” for the chicken soup product on a “yes/no/maybe” categorical scale. If consumers selected “yes”, they would rate their perceptions about “natural” on the same categorical scale; otherwise, they were required to indicate which ingredient(s) made them consider that the soup product was or might be “artificial”. In addition, consumers’ interest in consuming the chicken soup was evaluated on a five-point scale (1—definitely would not consume; 3—might or might not consume; 5—definitely would consume).

#### 2.3.3. Kano Analysis

There were 14 features about chicken soup products that were tested (Appendix A). Majority of them were from claims of commercial products and from the results of our previous study [7]. Consumers were asked to answer three questions for each feature. The first question was a functional form, which captured the consumer’s response to the presence of the attribute. The second question was a dysfunctional form that captured the consumer’s response if that attribute was absent. A five-point scale was applied with labels in the order of “I will like it”, “I must have it”, “I don’t care”, “I can live with it” and “I will dislike it” from left to right. The third question was a nine-point self-stated importance scale (1—not at all important; 3—somewhat important; 5—important; 7—very important; 9—extremely important).

#### 2.3.4. Demographic and Behavior Questionnaire

In the last part of the questionnaire, demographic data were collected. Consumer’s concern level about MSG was evaluated on a seven-point scale [7]. In addition, some consumption and purchase questions about chicken soup products were asked.

### 2.4. Statistical analysis

The data from DODC and consumer tasting test were analyzed by two-way and one-way analysis of variance (ANOVA) respectively in SAS (version 9.4, SAS Institute, Cary, NC, USA) using the GLIMMIX procedure (General Linear Mixed Models). Panelist was considered as a random factor. Least square means were calculated. Post-hoc mean separation was done using Fisher’s least significant difference (*P* < 0.05). JAR data were grouped to three categories that were “too little” (<4), JAR (4), and “too much” (>4). Their percentage distributions were reported. One-sample *t*-test was then conducted in SAS to compare individual product’s mean JAR intensity versus the ideal JAR intensity which was 4 on the scale.

For Kano analysis, this study used the continuous analysis method proposed by Bill DuMouchel, which solved the challenges in traditional discrete analysis method like misclassification caused by similar counts for several categories and incapability to distinguish the features under the same category [17]. The options of Kano scale were translated to a numeric value as follows:

Functional (Y): −2 (Dislike), −1 (Live with), 0 (Don’t care), 2 (Must-have), 4 (Like);

Dysfunctional (X): −2 (Like), −1 (Must-have), 0 (Don’t care), 2 (Live with), 4 (Dislike).

For each question, the averages of X (dysfunctional), Y (functional), and importance score were computed across the respondents. Only the averages that fall in the range of 0 to 4 were displayed on a scatter plot. The plot was divided into quadrants corresponding to one-dimensional, must-have, attractive, and indifferent (Figure 1). Then, the average of the importance score for each question was visualized by converting its scatter plot dot to bubble with size proportional to its importance.

## 3. Results and Discussion

### 3.1. Degree of Difference from Control (DODC) Testing

Eight attributes were measured in this test and significant differences (*P* < 0.05) were detected for seven of them (Table 3). Only yeast flavor, the identity (ID) flavor of YE, was statistically same (*P* ≥ 0.05) among six chicken soup samples. The mean degree-of-difference-from-control (DODC) intensities of the eight attributes for all the samples are presented in Table 3. The “Control” sample was compared against the “Control” reference and its DODC intensities were “zero” for all the attributes, which implied the accuracy of the descriptive panel. Addition of individual flavor enhancer (MC 0.1%, MSG 0.1%, TC 0.2%, YE 0.025%) significantly enhanced the overall flavor, meaty flavor, saltiness, and umami taste of the chicken soup added with 0.4% salt. YE 0.025% showed statistically same enhancement effect as MSG 0.1% in all the attributes. MC 0.1% sample exhibited a significantly weaker improvement in the chicken flavor and umami taste than MSG 0.1% but had a stronger mushroom ID flavor. This meant that higher concentrations of MC would contribute more mushroom flavor to the soup instead of enhancing the chicken flavor. This phenomenon was also observed with TC. TC was used at the highest level among flavor enhancers which successfully boosted the overall flavor, meaty flavor, salty taste, and umami taste of chicken soup to the level achieved by MSG 0.1%. However, it also significantly increased the tomato ID flavor. In addition, TC 0.2% sample had a significantly lower chicken flavor than MSG 0.1% and YE 0.025% and lower salty taste than YE 0.025%. In addition to glutamate, YE’s higher nucleotide content might be the reason for its high umami perceptions and enhancement effects (data submitted elsewhere). Jo and Lee came to similar conclusions as ours while comparing commercial yeast extracts and their combinations with MSG in water solutions [9].

Salt is crucial to the palatability of a wide variety of foods. Besides increasing the saltiness, salt also contributes to the overall flavor and suppression of bitter taste [18]. In this study, Salt 0.5% sample had a significantly higher (*P* < 0.05) DODC value of most attributes than Control (0.4% salt sample), suggesting that usage of extra 0.1% of salt improved the flavors and tastes of chicken soup. On the other hand, 20% salt reduction was already able to significantly impact the sensory profile of chicken soup. It is well known that umami taste enhances saltiness and other flavor attributes. Previous research studies have reported that addition of yeast extract could reduce sodium usage without sacrificing sensory pleasantness [12,19]. Our results indicated that YE 0.025% and MSG 0.1% had similar effects as Salt 0.5% on overall flavor and salty taste. Compared to Salt 0.5% sample, our panelists perceived significantly stronger meaty flavor, chicken flavor, and umami taste in YE sample. Although MC 0.1% and TC 0.2% samples had significantly lower (*P* < 0.05) salty taste than Salt 0.5% sample, they still showed a similar or even higher intensity in majority of the attributes. Overall, our findings proved that all the four flavor enhancers at current level could compensate for the flavor and taste loss caused by 20% salt reduction.

### 3.2. Consumer Taste Test

A total of 93 consumers participated in the test. The mean intensity of three hedonic responses were significantly different among six chicken soup samples (Figure 1). Same liking pattern was observed for overall liking and overall flavor liking. MSG 0.1% had highest score for these two hedonic attributes, followed by YE 0.025%. In general, consumers slightly liked MSG 0.1%, YE 0.025%, and Salt 0.5% sample with no statistical difference among them. They also obtained significantly higher (*P* < 0.05) acceptability scores than Control, MC 0.1%, and TC 0.2%. The capability of MSG in sensory enhancement has been widely proved in different food product. Ribonucleotides (mainly IMP and GMP) also contribute to palatability elevation, which exhibit synergistic influences with MSG. Miyaki et al. reported that the overall liking of chicken noodle soup was significantly increased (*P* < 0.05) by using 0.1% MSG or 0.3% MSG + 0.1% IMP, and the sample enhanced by IMP and MSG together was liked most among all the samples [6]. Baryłko-Pikielna and Kostyra studied the effects of MSG (0–0.5%) and IMP +GMP (0–0.015%) on the palatability of seven model food matrices [4]. Application of MSG and/or IMP+GMP increased the hedonic response in all model products, but the degree of elevation varied considerably among the products. In their work, MSG played a leading role in the increased acceptability of chicken soup. Yeast extract is rich in both glutamate and ribonucleotides. Therefore, it was not surprising to find that compared to MSG, YE increased the palatability of chicken soup to a similar level at a much lower dose. Mushrooms and tomatoes are also very rich in umami substances. However, their extracts failed to significantly increase the overall liking and overall flavor liking of the chicken soup. When cross-compared to the DODC findings, although MC 0.1% and TC 0.2% sample had significantly higher values for all the sensory attributes than Control, they still showed a significantly lower chicken flavor than MSG 0.1% and YE 0.025%, and lower saltiness than Salt 0.5% and YE 0.025%. Therefore, a relatively lower chicken flavor and salty taste might have caused their failure in acceptability improvement. Previous studies have also reported a positive relationship between chicken flavor and overall liking of chicken meat and stock products [20,21]. In addition, TC 0.2% had a DODC intensity of 2.5 for tomato flavor which indicated a slight difference from Control. This slightly higher tomato flavor could also adversely affect consumer acceptability of chicken soup. Top two boxes score (combined percentage for the responses of “like extremely” and “like very much”) for overall liking represents the percentage of consumers that strongly liked the products. Its findings were generally in an agreement with liking means (data not presented). More than 20% of participants generally liked Salt 0.5% (25.8%), MSG 0.1% (23.7%), and YE 0.025% sample (23.7%). The top 2 box percentage for Control and MC 0.1% was 14.0% and 16.1% respectively. Only 8.6% of consumers liked TC 0.2% extremely or very much. In contrast to the Control, all the enhanced chicken soup samples had a significantly higher appearance liking. Consumer did not perceive any difference in the appearance liking among MC 0.1%, MSG 0.1%, Salt 0.5%, and YE 0.025%. All these four samples were liked significantly more than TC 0.2% appearance-wise. The difference in appearance likings might origin from the Halo effect of overall liking. Overall, palatability of chicken soup could be improved by adding 0.1% MSG, 0.025% YE, or extra 0.1% salt. This increased acceptability might result from increased chicken flavor and/or salty taste.

Consumers also evaluated the optimal intensity of color, overall flavor, salty, and savory (umami) tastes using seven-point JAR scale (Figure 2a–d). Samples that were liked more by consumers also got a larger percentage for JAR category (4 on the scale). Similar distribution pattern of JAR data was observed among overall flavor, salty, and savory tastes of six chicken soups. YE 0.025%, MSG 0.1%, and Salt 0.5% consistently had lesser “too low” and more JAR scores than other samples for these three attributes. The pattern for color JAR was slightly different because the scores for MC 0.1% were almost evenly distributed among three categories. Eighty percent of JAR scores is considered as a common benchmark by some product developers [22]. In this study, most of the JAR scores were less than 50%, which was far below the benchmark of 80%. One-sample *t*-test was also conducted to compare individual product’s mean JAR intensity versus the center point’s value on the scale (4) for each attribute. A significantly lower value was detected for all the tests, indicating the mean intensity was very much skewed below the appropriate level. 

Despite all the samples not being close to the ideal, the enhancement effect was still noticeable. Control sample had smallest percentage of JAR for color, salty and savory tastes. Using either flavor enhancers or extra salt successfully increased the JAR scores of these three attributes but to different extents. YE 0.025% showed the largest improvement especially in the overall flavor and savory taste. Approximate two-fold increment in JAR percentage was achieved by addition of YE at 0.025% in contrast to Control. Salt 0.5% and MSG 0.1% had nearly same proportion of JAR for color, salty, and savory tastes respectively. TC 0.2% and MC 0.1% showed an extremely weak influence on the JAR of savory taste when compared to Control. As for JAR of overall flavor, using TC at 0.2% slightly reduced the percentage of appropriateness compared to Control. In DODC testing, TC 0.2% sample had stronger overall flavor than Control and no difference in this attribute from other enhanced samples. However, lowest proportion of consumers voted JAR for overall flavor of TC 0.2%, which was about half of that for YE 0.025%. The inconsistent findings of overall flavor in consumer and DODC study possibly result from different understandings of overall flavor among consumers and trained panelists. 

### 3.3. Effect of Information on Consumer Perceptions

Food packaging and labelling are normally the first contact between consumer and a processed food product [14]. Since consumer’s ingredient awareness for natural and healthy foods rise, food label plays a significant role in consumer food choices. As an important component of food label, an ingredient panel provides the information that might affect consumer perception about “naturalness”. In this study, ingredient lists for the chicken soup samples used in the taste test were investigated among 93 consumers. Since Control and Salt 0.5% had the same ingredient information, only five lists were applied. Consumers firstly evaluated their perceptions about “free of artificial ingredients” for individual ingredient panel. If they considered the panel had or might have artificial ingredients, they were required to select all the ingredients that gave them the artificial perception. Table 4 shows that only 31.2% of participants perceived that Panel A (MSG) did not contain artificial ingredient, which was lowest. Another 47.3% considered MSG-contained chicken soup had artificial ingredients and MSG was the only ingredient selected by them for artificial perception. The rest of 21.5% of consumers thought Panel A might have artificial ingredients and 85% of them voted for MSG (data not presented). By changing MSG to its substitutes, more consumers considered the product as free of artificial ingredients. The degree of elevation followed the ascending order of Panel D (YE) < Panel B (MC) < Panel C (TC). The exact same order was observed in our previous study, which surveyed consumer’s choice of MSG substitutes in chicken soup [7]. There were around 10–27% of consumers still considered the chicken soup with MSG substitutes were or might be artificial, and majority of them pointed to the corresponding alternative in the ingredient panel. Panel E (Salt) which did not include any flavor enhancer obtained highest percentage (91.4%) for “free of artificial”. Previous studies found that consumers perceived less naturalness from food additive with hard-to-pronounce name, chemical name, or from less familiar source. Thus, the chemical name of MSG resulted in the highest responses for “artificial”. In addition, consumers might be more familiar with tomato and mushroom in food, leading to a higher percentage of “artificial-free” than YE. Rozin indicated that using additives is associated with “chemical transformation”, which destroyed the naturalness more compared to physical changes like grinding and freezing [23]. This likely explains why more participants considered the enhancer-free ingredient panel as free of artificial ingredients.

The perception of naturalness was further explored for the consumers who considered the product did not have any artificial ingredient. Similar to “free of artificial ingredient” perception, the chicken soup products considered least and most natural was Panel A (MSG) and Panel E (salt) correspondingly (Table 4). According to our previous study, the reasons for artificial image of MSG included “not present naturally in food”, “its name and/or its abbreviation”, and “manufactured from inedible materials/chemicals” [7]. This, on the other hand, explained why chicken soup contained MSG substitute had higher natural perception. Other scientists also reported that only 2% respondents considered MSG natural and concluded that consumers did not know what ingredient was and was not natural with various reasons to justify their natural perception, which introduced significant difficulties for FDA to come up with an acceptable definition agreed by consumer, academia, and producer [24]. As shown in Table 4, the percentage for “natural” was consistently less than that for “free of artificial” for individual ingredient panel. This implies that “free of artificial ingredients” might only be a component of consumer “natural perception”. 

Consumers also rated their interest in consuming the chicken soup for corresponding ingredient panel. Consumers had the strongest interest in the flavor enhancer–free product, which was significantly higher (*P* < 0.05) than the products containing enhancer (Table 4). Among the enhancers, consumers possessed a significantly higher (*P* < 0.05) interest in TC and MC-enhanced product. YE containing product had a value, which was significantly lower than the rest. A mismatch was noticed when comparing the results of taste test to label perception test. YE- and MSG-added chicken soup, which consumers liked most in taste test obtained the lowest interest in label perception test. This discrepancy also existed in TC and MC sample. When consumers did not taste the chicken soup, their interest in trying the product seemed to correlate to their perceptions of naturalness and free-of-artificial-ingredient about the corresponding ingredient panel. The decision about everyday food choice is a trade-off between various sensory and non-sensory factors [13]. The effects of information on consumer perceptions about food products have been investigated by many researchers. Bandara et al. reported that list of ingredients was ranked as the second most important mandatory labeling information by consumers and only 3% respondents’ purchase intent was not influenced by MSG content [15]. Besides ingredient, family influence, price, and taste also drive consumer attitudes towards food products [25]. It is likely that consumers would imagine the taste of products by looking at the information on the package [14]. We speculate that the unpleasant taste imagined by consumers for yeast extract might result in the lowest interest in consuming the corresponding chicken soup product. Prescott and Young found that the information about MSG usage did not influence consumer liking, natural taste perception, and purchase intent of vegetable soup when the product and label information were served together to consumers [16]. They claimed that when tasting the products, consumers weighted sensory properties more important than information highly relevant to their beliefs and attitudes. Due to the financial and time limitation, we did not test the effect of ingredient information on hedonic responses. However, based on previous study, information about taste might increase consumer willingness to try food products [26].

Considering that the habit of reading ingredient panel during shopping for chicken soup products might affect consumer perceptions about the ingredient information, consumers were further separated to two groups. The first group contained 56 consumers who read or sometimes read ingredient panel of chicken soup products during shopping; the other 37 consumers did not have this shopping habit. The percentages of selection for “free of artificial ingredients” and “natural” were calculated for each ingredient panel, no difference was noted among two groups (data not presented). ANOVA was also conducted to analyze the differences in the interest of consumption between the two groups. No significant group effect or group by ingredient effect was detected (data not presented). These findings indicated that whether consumer read ingredient list or not they had no difference in free-of-artificial and natural perception and consumption interest in in chicken soup products. 

### 3.4. Kano Analysis

Kano questions are used to understand the attributes affecting consumer satisfaction or dissatisfaction about a product. There are six categories for product features in the Kano model [17]. Must-have feature is the attribute expected by customers and sometimes taken for granted when fulfilled. Its absence or poorly satisfaction results in large customer dissatisfaction. One-dimensional feature has a positive linear relationship to customer’s satisfaction. Attractive feature provides great satisfaction when fulfilled but are acceptable when not fulfilled. Indifferent attribute does not affect consumer satisfaction. Reverse feature results in dissatisfaction when fulfilled and in satisfaction when not fulfilled. When a consumer has conflicting responses, the attribute goes to the category of questionable feature. 

In this study, none of the 14 attributes belonged to questionable or reserve category. “Taste good” was the only one-dimensional attribute for chicken soup product (Figure 3). Tasting good is a must-have attribute for bacon but a one-dimensional attribute for cottage cheese [27,28]. “Free of artificial ingredient”, “free of added MSG”, “all natural”, “free of preservative”, “ready to serve”, and “lower price” were attractive attributes for chicken soup products. Most of these attractive attributes were relevant to food additives. The rest of the seven attributes were all in the group of indifferent attributes. Similar as our study, consumer also indicated “organic”, “low sodium” and “fat-free” were indifferent features for cottage cheese [28]. As an important category of chicken soup products, it was surprising to find that “low sodium” was indifferent to consumers. It seems that this group of consumers cared more about artificial or natural ingredient usage than sodium reduction in chicken soups. Both ready-to-serve and on-the-go packaging are convenience-related features. Consumers considered “ready-to-serve” attractive but “on-the-go” packaging indifferent. Normally, feature classification transitions in the direction of indifferent to attractive to one-dimensional and finally to must-have due to market maturity [29]. Since “on-the-go” is a relatively new concept in chicken soup products, it is likely that consumers might value it more as its growth in the market.

The self-stated importance rating was firstly compared among all attributes using ANOVA and then projected on to the Kano plot with the size of individual attributes proportional to its importance. “Taste good” was the most important attribute to consumers with a score of 8.1 on a 9-point scale, which was significantly higher (*P* < 0.05) than all the other attributes (data not presented). “Free of added MSG” was the second important attribute of chicken soup products. Consumer perceived no difference in the importance of “free of added MSG”, “all natural”, “lower price”, “GMO free”, “free of artificial ingredients” and “free of preservatives.” But only “free of added MSG” obtained a score of 5 which was the benchmark for important attributes on a 9-point scale. Therefore, “Free of added MSG” was the only attractive feature that consumers felt important. According to FDA, foods with any ingredient that naturally contains MSG cannot claim “No MSG” or “No added MSG” on their packaging [30]. However, there are some manufacturers still claiming “No (added) MSG” on either packaging, website, or other sources for soup products enhanced by MSG substitutes that naturally contain MSG. More clear definition and stronger regulation may be required to clarify MSG-related claims. According to Figure 3, the self-stated importance generally decreased in the order of one-dimensional > attractive > indifferent. However, GMO-free, an indifferent feature, obtained an importance score that was statistically same to most of attractive attributes. “Ready-to-serve” which was attractive to consumers grouped with other indifferent attributes from viewpoint of self-stated importance.

### 3.5. Influence of MSG Concern Level on Responses

Consumers also indicated their concern level about MSG usage in food products on a seven-point Likert-like scale. Based on their responses, consumers were clustered into three groups: the first group was the concerned group which consisted of 43 consumers who rated their concern level at 5 and above; the second group was the unconcerned group which had 33 consumers whose concern level was below 5; and the rest of 17 consumers did not hear about MSG before, so they were considered as a separate group to avoid bias. The average concern level for the first group was 5.7, which was significantly higher than that of the second group (3.2), and third group (4.1). Consumers in the third group were also concerned more than consumers in the second group. These three groups were then compared for any difference in the other responses. No difference among three groups was detected for overall liking of the six chicken soup samples and perceptions about the individual ingredient panel. As for Kano analyses, most of findings aligned with previous results. “Free of preservatives” became one-dimensional feature to consumers in concern group. Consumers who did not hear about MSG perceived “added nutrients”, “free range chicken” and “on-the-go packaging” significantly more important than consumer in concern group did.

### 3.6. Soup Eating and Purchasing Behavior Questions

Behavior questions about chicken soup products were also investigated in this work. Consumers were firstly required to select top three health benefits that they would like to see most in chicken soup products. Higher protein, added vegetables, and low sodium were the top three choices with 69.9%, 62.4%, and 49.5% respectively. Other benefits included low calorie (41.9%), no trans-fat (38.7%), added vitamins (24.7%), and added minerals (12.9%). It seems that using more vegetables in chicken soup could be a good strategy to increase consumer intake of vitamins and minerals. Consumers who selected low sodium in previous question were further asked about the level of sodium reduction they wanted most in chicken soup. Results showed that 43.5% liked 30–50% sodium reduction and 26.1% voted for less than 30% sodium reduction. Thus, majority of participants preferred 50% and less sodium reduction in chicken soup products. As for the reasons or motivations for chicken soup consumption, 68.8% expressed that they ate chicken soups when they are sick. Around 63.4% considered chicken soup easy and fast to prepare; another popular reason was the affordability (44.1%) of chicken soups. Other motivations and reasons were “personal preference” (38.7%), “maintain a healthy diet” (36.6%), and “family tradition” (24.7%). Ready-to-eat was the type of chicken soup products that most consumers (80.6%) usually purchased. Condensed (67.7%) and wet broth/stock (65.6%) also had more than 60% selection. Dry and frozen/refrigerated chicken soup products were less popular among this group of consumers with a percentage of 29% and 10.8% respectively.

## 4. Conclusions

Overall, MSG and all of its alternatives enhanced the flavors and tastes of chicken soup with 0.4% salt. YE 0.025% had the same effects as MSG 0.1% in all tested sensory attributes. All the flavor-enhanced samples had the same or even stronger overall flavor, meaty flavor, chicken flavor, and umami taste when compared to chicken soup added with 0.5% salt. But addition of MC 0.1% and TC 0.2% failed to increase the salty taste to the level achieved by using 0.5% salt. Consumers significantly preferred MSG 0.1%, YE 0.025%, and Salt 0.5% samples to the others. Information about ingredient usage largely affected consumer perceptions about the chicken soup. Our results showed that familiarity bias toward ingredient name played a significant role in consumer “free of artificial” and “natural” perception and interest to consume chicken soup products. MSG and yeast enhanced sample which had higher consumer likings in taste test obtained a lower consumer interest in ingredient panel testing. Kano analysis indicated that “good taste” was the most important feature influencing consumer satisfaction about chicken soups. Claims related with artificial and natural ingredient were attractive attributes to consumers. Our findings suggested that using natural extracts, especially yeast extract, could successfully compensate for the palatability loss caused by removal of MSG or 20% salt reduction. However, there was a mismatch between consumer sensory liking and their expectations based on ingredient usage. More studies are required to investigate how information about ingredient usage changes consumer hedonic responses towards chicken soup products.

## Figures and Tables

**Figure 1 foods-08-00071-f001:**
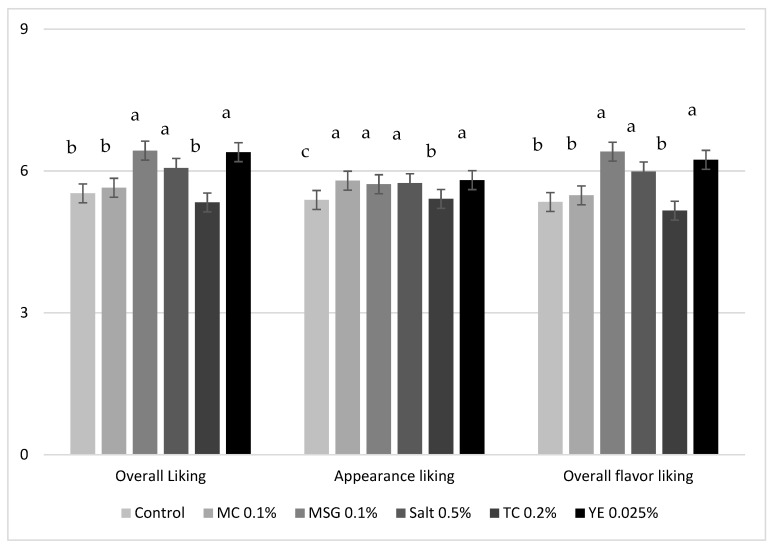
Mean consumer liking scores for chicken soups. MC, mushroom concentrate; MSG, monosodium glutamate; TC, tomato concentrate; YE, yeast extract.

**Figure 2 foods-08-00071-f002:**
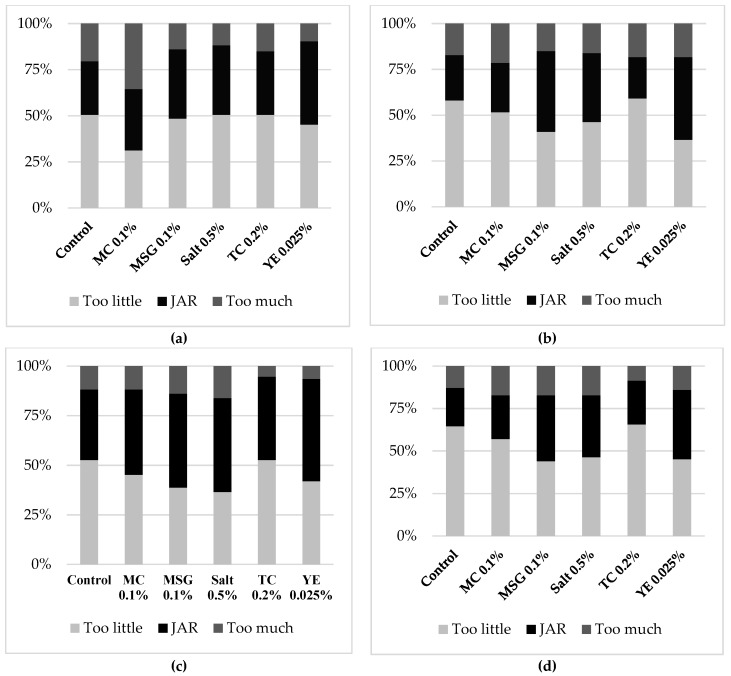
Distribution of consumer JAR scores for sensory attributes of chicken soups: (**a**) Color; (**b**) Overall flavor; (**c**) Salty taste; (**d**) Savory taste.

**Figure 3 foods-08-00071-f003:**
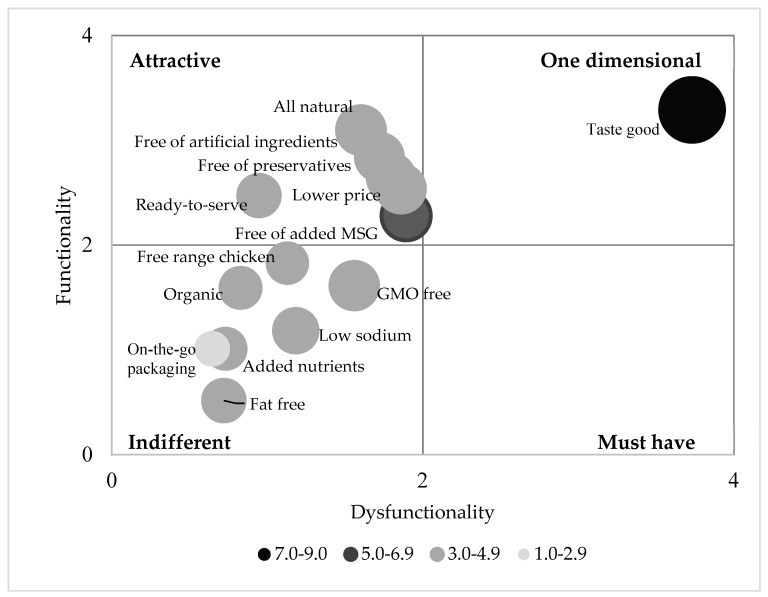
Kano classifications for 14 features about chicken soup products with averaged importance for each attribute indicated by size and color of the bubble.

**Table 1 foods-08-00071-t001:** Percentage (*w/v*) of salt (NaCl) and flavor enhancer in 1.5 L chicken soup.

Sample	Flavor Enhancer		Salt (%)
	Name	Level (%)	
Control	-	-	0.4%
MC 0.1%	Mushroom concentrate	0.1%	0.4%
MSG 0.1%	Monosodium glutamate	0.1%	0.4%
Salt 0.5%	-	-	0.5%
TC 0.2%	Tomato concentrate	0.2%	0.4%
YE 0.025%	Yeast extract	0.025%	0.4%

MC, mushroom concentrate; MSG, monosodium glutamate; TC, tomato concentrate; YE, yeast extract.

**Table 2 foods-08-00071-t002:** Ingredient lists of chicken soup samples used in consumer testing.

Panel	Abbreviation	Ingredient Panel
A	MSG	Water, chicken, carrot, onion, celery, salt, monosodium glutamate, bay leaf
B	MC	Water, chicken, carrot, onion, celery, salt, mushroom concentrate, bay leaf
C	TC	Water, chicken, carrot, onion, celery, salt, tomato concentrate, bay leaf
D	YE	Water, chicken, carrot, onion, celery, salt, yeast extract, bay leaf
E	Salt	Water, chicken, carrot, onion, celery, salt, bay leaf

**Table 3 foods-08-00071-t003:** The degree-of-difference-from-control (DODC) mean intensity scores (standard deviation in parentheses) of chicken soup samples on a 10-point intensity scale.

Soup	Overall Flavor	Meaty Flavor	Chicken Flavor	Salty Taste	Umami Taste	Yeast Flavor	Mushroom Flavor	Tomato Flavor
Control ^1^	0 (0) ^b,2^	0 (0) ^c^	0 (0) ^c^	0 (0) ^c^	0 (0) ^d^	0 (0)	0 (0) ^b^	0 (0) ^c^
MC 0.1% ^1^	3.1 (0.4) ^a^	3.0 (0.4) ^a^	0.8 (1.3) ^bc^	2.5 (0.6) ^b^	2.6 (0.6) ^bc^	0.1 (0.3)	1.2 (1.3) ^a^	0.1 (0.3) ^c^
MSG 0.1% ^1^	3.2 (0.4) ^a^	2.6 (0.7) ^ab^	2.2 (0.7) ^a^	2.9 (0.4) ^ab^	3.1 (0.4) ^a^	0.3 (0.7)	0.4 (0.7) ^b^	0.5 (0.7) ^b^
Salt 0.5%	3.3 (0.2) ^a^	2.2 (1.1) ^b^	0.8 (1.1) ^bc^	3.3 (0.3) ^a^	2.4 (0.9) ^c^	0.2 (0.5)	0.4 (0.6) ^b^	0.1 (0.3) ^c^
TC 0.2% ^1^	3.1 (0.4) ^a^	2.6 (0.7) ^ab^	1.2 (1.4) ^b^	2.5 (0.8) ^b^	2.8 (0.3) ^ab^	0 (0)	0.3 (0.5) ^b^	2.5 (0.5) ^a^
YE 0.025% ^1^	3.3 (0.4) ^a^	3.0 (0.4) ^a^	2.1 (0.8) ^a^	3.1 (0.5) ^a^	3.0 (0.4) ^ab^	0.6 (1.4)	0.1 (0.3) ^b^	0.3(0.6) ^bc^

^1^ Contained 0.4% salt; ^2^ Different letters in the same column indicates significant difference (*P* < 0.05).

**Table 4 foods-08-00071-t004:** Consumer perception about the five ingredient panels for chicken soup products.

Panel ^1^	Abbreviation	Free of Artificial			Natural ^2^			Interest Intensity for Consumption ^3^
		Yes	No	Maybe	Yes	No	Maybe	
A	MSG	31.2%	47.3%	21.5%	18.3%	9.7%	3.2%	3.7 ^c,4^
B	MC	65.6%	20.4%	14.0%	50.6%	7.5%	7.5%	4 ^b^
C	TC	76.3%	9.7%	14.0%	62.3%	6.5%	7.5%	3.9 ^b^
D	YE	50.5%	22.6%	26.9%	36.5%	7.5%	6.5%	3.5 ^d^
E	Salt	91.4%	6.5%	2.2%	87.0%	2.2%	2.2%	4.5 ^a^

^1^ Refer to Table 3 for the detailed information about the ingredient panel; ^2^ Only consumer who selected “Yes” for “Free of artificial ingredient” answered this question; ^3^ Consumer rated their level of interest to consumer on a 5-point Likert-like intensity scale; ^4^ Different letters in the same column indicates significant difference (*P* < 0.05).

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
