# Peer review of "Influence of Monosodium Glutamate and Its Substitutes on Sensory Characteristics and Consumer Perceptions of Chicken Soup"

_foods, 2019, doi:10.3390/foods8020071_

Reviewer 1 Report

The objective of this research paper was to evaluate the effects of monosodium glutamate (MSG) and its substitutes on the sensory characteristics and consumers perceptions of chicken soup.  This study also evaluates the reduction of sodium and label claims on consumers acceptability and purchase behaviour.  This paper is well-written, and the statistical analysis is sound. However, there are several points that need to be clarified before publication (mainly in the materials and methods). More specific comments can be found below:

Abstract: A statement indicating the reason behind substituting MSG in food products should be incorporated at the beginning of the abstract. Besides, the abstract should mention the names of the MSG substitutes used in this study when mentioning the methodology.

Materials and Methods:

Table 1: This table shows the formulation for each treatment.  It is unclear how these percentages were selected for the study.  The methods should clearly explain the rationale behind selecting 0.1% for mushroom concentrate and MSG, 0.2% for tomato concentrate, and only 0.025% for yeast concentrate.

Line 104-105: The process of adding the salt and/or flavour enhancer is not explained. The process of mixing or homogenization of salt/flavour with the stock solution should be mentioned.

Line 107-126: The description of the training sessions needs more information. Were references/standards used in the familiarization process? Were panellists trained in using the different scales for the sensory evaluation?  This information is important because the results show the comparison of the different treatments using the panellists scores.  

Line 107-126: Indicate how many samples were served in a single session. From the description, it seems that 20 samples were served (10 pairs). Is this correct?

Line 144-151: Explain why the labels were presented using a fixed order and not random.  Was the purpose of this to compare the other labels with the MSG label?  Even if labels are considered visual stimuli, there is a possibility of having an order effect if randomization is not used.

 Line 153-161: The information of the 14 attributes should be added in the description. Besides, it should be mentioned how these attributes were selected.

Results:

Table 3: Add the standard deviation for each mean.

Line 187-220: From the results, it seems that yeast extract was the closest to MSG in terms of the sensory attributes.  Since this paper is about describing the substitutes of MSG, it is important to add more discussion about why the yeast extract is similar to MSG, and why the other substitutes did not achieve those similar results. Moreover, a question remains whether adding a higher concentration of the other substitutes (tomato, mushroom) in the soup could possibly act similarly to MSG with some other drawback effect in terms of the sensory perception.

Line 257-260: The aroma of the soup could also impart a Halo effect to the appearance scores.  Was aroma considered for this sensory evaluation?

Figure 1: Add error bars to this figure.

Line 263-295: Instead of votes, use scores. Check this for all the manuscript.

Table 4: Frequencies, proportions, or percentages can be statistically compared using the Cochran Q test.  Doing this analysis can improve the validity of these results.

Author Response

The objective of this research paper was to evaluate the effects of monosodium glutamate (MSG) and its substitutes on the sensory characteristics and consumers perceptions of chicken soup.  This study also evaluates the reduction of sodium and label claims on consumers acceptability and purchase behaviour.  This paper is well-written, and the statistical analysis is sound. However, there are several points that need to be clarified before publication (mainly in the materials and methods). More specific comments can be found below:

Abstract: A statement indicating the reason behind substituting MSG in food products should be incorporated at the beginning of the abstract. Besides, the abstract should mention the names of the MSG substitutes used in this study when mentioning the methodology.

 The statement and names of substitutes are added in the abstract – Line 10-13

Materials and Methods:

Table 1: This table shows the formulation for each treatment.  It is unclear how these percentages were selected for the study.  The methods should clearly explain the rationale behind selecting 0.1% for mushroom concentrate and MSG, 0.2% for tomato concentrate, and only 0.025% for yeast concentrate.

The results for selection of the usage levels are written in our manuscript which is under review at Journal of food science. Generally speaking, we selected the usage level by 1) comparing umami substances of each ingredient, which were quantified by high performance liquid analysis and determining the Equivalent Umami Concentration (EUC); 2) using trained panel to compare the similarity in sensory enhancements (4 levels applied at the beginning). The selected level for each ingredient had similar effects on sensory attributes of chicken soup, which were shown by the DODC results.

The reasons for a lower yeast extract level is its high nucleotide concentration which amplifies the umami perceptions and enhancement effects through synergisms with glutamate. The higher tomato level is due to its slightly weaker effects on chicken soup compared to MSG at the same level. It has been explained in line 105-108.

Line 104-105: The process of adding the salt and/or flavour enhancer is not explained. The process of mixing or homogenization of salt/flavour with the stock solution should be mentioned.

‘The ingredients were added to 1.5 L of the stock at the beginning of the soup preparation and completely dissolved in the stock by string.’ Added at line 107-108.

Line 107-126: The description of the training sessions needs more information. Were references/standards used in the familiarization process? Were panellists trained in using the different scales for the sensory evaluation?  This information is important because the results show the comparison of the different treatments using the panellists scores.  

We have explained the method with more clarity – Lines 116-141.

Line 107-126: Indicate how many samples were served in a single session. From the description, it seems that 20 samples were served (10 pairs). Is this correct?

We had 6 testing samples (as shown in Table 1) and conducted 2 replications, thus there were 12 samples in total. Each test sample was served with ‘Control’ which can be considered reference for intensity evaluation – Lines 138-141

Line 144-151: Explain why the labels were presented using a fixed order and not random.  Was the purpose of this to compare the other labels with the MSG label?  Even if labels are considered visual stimuli, there is a possibility of having an order effect if randomization is not used.

We used Compusense for data collection. Each label panel had different information. Under each label panel, there were multiple questions and consumer were branched to each question according to their previous answers. We couldn’t find a feasible way to randomize the label order on Compusense given the situation because of the branching. Thus, a fixed order was used. As pointed out by you – since it was not taste stimuli, we thought that the order effect would be minimal.

Line 153-161: The information of the 14 attributes should be added in the description. Besides, it should be mentioned how these attributes were selected.

This has been provided as Appendix A. The attributes chosen were of regular concerns of consumers that were chosen from focus groups [7] and product labels – Lines 170-172.

Results:

Table 3: Add the standard deviation for each mean

 Standard deviation are presented in parentheses

Line 187-220: From the results, it seems that yeast extract was the closest to MSG in terms of the sensory attributes.  Since this paper is about describing the substitutes of MSG, it is important to add more discussion about why the yeast extract is similar to MSG, and why the other substitutes did not achieve those similar results. Moreover, a question remains whether adding a higher concentration of the other substitutes (tomato, mushroom) in the soup could possibly act similarly to MSG with some other drawback effect in terms of the sensory perception.

We have addressed this in Lines 215-224.  

Line 257-260: The aroma of the soup could also impart a Halo effect to the appearance scores.  Was aroma considered for this sensory evaluation?

MSG and its substitutes are used as flavor enhancer which aims to improve the flavor and tastes of products. Thus, our work mainly focused on these attributes rather than orthonasal aromas. Moreover, during training sessions, our panel didn’t detect any major aroma differences in chicken soup samples caused by MSG and substitutes, thus we decided to remove aromas totally from this study.

Figure 1: Add error bars to this figure

Added

Line 263-295: Instead of votes, use scores. Check this for all the manuscript.

Changed throughout the manuscript.

Table 4: Frequencies, proportions, or percentages can be statistically compared using the Cochran Q test.  Doing this analysis can improve the validity of these results.

The primary assumption for Cochran’s test is that the response should be binary (0, 1; success, failure) in nature. Cochran Q test will not work for our study because we used three categories (Yes/Mo/Maybe).

https://ncss-wpengine.netdna-ssl.com/wp-content/themes/ncss/pdf/Procedures/NCSS/Cochrans_Q_Test.pdf

Reviewer 2 Report

The paper by Wang, Zhang and Adhikari, aimed to study the influence of monosodium glutamate and its, less controversial, substitutes on chicken soup sensory characteristics and consumer perceptions. The specific comments are as follows:

Materials and Methods

L100: “…64 oz. glass jar…” Consider using uniform units.

L106: Why authors considered these inclusion levels?

L142: Consider changing “…was used for intensity questions.” to “…was used for color, overall flavor, salty, and savory (umami) tastes intensity questions.”

Author Response

Reviewer 2

The paper by Wang, Zhang and Adhikari, aimed to study the influence of monosodium glutamate and its, less controversial, substitutes on chicken soup sensory characteristics and consumer perceptions. The specific comments are as follows:

Materials and Methods

L100: “…64 oz. glass jar…” Consider using uniform units.

Changed to 1.89 L

 L106: Why authors considered these inclusion levels?

The results for selection of the usage levels are written in our manuscript which is under review at Journal of food science. Generally speaking, we selected the usage level by 1) comparing umami substances of each ingredient, which were quantified by high performance liquid analysis and determining the Equivalent Umami Concentration (EUC); 2) using trained panel to compare the similarity in sensory enhancements (4 levels applied at the beginning). The selected level for each ingredient had similar effects on sensory attributes of chicken soup, which were shown by the DODC results.

 L142: Consider changing “…was used for intensity questions.” to “…was used for color, overall flavor, salty, and savory (umami) tastes intensity questions.”

Changed as suggested – Lines 157-158.